# Influence of Appeal Type and Message Framing on Residents’ Intent to Engage in Pro-Environmental Behavior

**DOI:** 10.3390/ijerph192315431

**Published:** 2022-11-22

**Authors:** Jia Wang, Yangli Gu, Haohang Xin, Xiaomei Wang

**Affiliations:** School of Media Studies and Humanities, Zhejiang University City College, Hangzhou 310015, China

**Keywords:** message framing, appeal type, waste separation, proenvironmental identity, environmental risk perception, proenvironmental behavior

## Abstract

In this intervention study, we examined the effects and psychological processes of different types of messaging intended to promote waste-separation behavior on the intent of residents to separate household waste and to engage in other pro-environmental behaviors (PEBs). Based on three studies (one survey with 76 residents and two experiments with 446 residents) in China, our results revealed that environmental appeals increased residents’ intent to separate waste and engage in other PEBs, whereas monetary incentives had no effect. Although the main effects of the framing of messages on the intent to separate waste and other PEBs were not significant, we found an interactive effect of appeal type and message framing. When matched with loss framing, an environmental appeal indirectly generated a stronger positive impact by stimulating an individual’s environmental risk perception. When monetary incentives matched with loss framing, individual pro-environmental identity weakened, thereby inhibiting the positive impact on intent to separate waste. With this study, we provide important theoretical support for the selection and optimization of waste-separation intervention strategies and guidance for persuading residents to engage in more PEBs.

## 1. Introduction

The development of urbanization and industrialization has had many negative impacts on ecosystems and human health, most of which stem from the imbalance between human social development and ecological governance [1]. The latest Global Risks Report released by the World Economic Forum 2022 pointed out that in the next 5–10 years, environmental risks such as climate action failure, extreme weather, and biodiversity loss rank as the top three most severe risks [2]. Serious ecological crises have aroused widespread public concern and prompted researchers to investigate the mechanisms of individual pro-environmental behaviors (PEBs) to provide practical strategies for improving environmental conditions [3].

PEBs are generally referred to as “environment protective” or environmentally friendly behavior and involve human activities performed with the aim of protecting the environment or preventing environmental degradation [4], for example, green travel, green consumption, and waste separation. As a major form of public participation in environmental governance, PEBs can significantly improve environmental quality while also providing residents with a sense of wellbeing and life satisfaction [5]. Whereas the specific PEBs vary widely in terms of residents’ effort, relative financial cost, knowledge, and a range of other factors, some research has shown that there may be some consistency across individuals’ different PEBs, and one PEB may affect the other [6,7]. For example, those who reduced their red meat consumption were more likely to make donations to environmental organizations [8]. In addition, the likelihood that an individual will adopt one or more non-targeted PEBs depends on the mode of intervention for the target PEB. Evans et al. (2013) found that discarded paper recycling was significantly higher than control when participants received environmental information about car-sharing, but was no different from control when they received financial information or received both financial and environmental information [9].

Among the many PEBs, the level of waste separation is widely recognized as a key indicator for measuring a community’s quality of life and urban sustainability. A report released by the World Bank in 2018 showed that in high-income countries, one-third of the waste is recycled and composted, whereas in low-income countries, the recycling rate is only 4% [10]. China is one of the countries with the highest pressure for municipal domestic waste disposal in the world, with domestic waste production increasing dramatically at a rate of 8–10% per year [11]. However, the recycling rate of domestic waste in China was only 15.6% in 2015 [12]. Given the urgency and importance of waste-separation management, the State Council promulgated the “Implementation Plan of the Household Waste-Separation System” in 2017. Grassroots governments and communities have adopted a variety of behavioral interventions such as publicity, education, and monetary incentives to encourage people to separate waste, and waste separation has gradually become one of the most common types of PEBs practiced by Chinese residents. Consequently, in 2022, China’s waste-recycling rate has increased to 35–40% [13].

According to Thøgersen and Crompton [7], as a simple and easy PEB in the private sector, waste separation can be a “wedge” or catalytic behavior that promotes other environmentally related behaviors. However, in the current literature on waste separation, academic experts have mostly focused on factors influencing public psychology, decision-making, and behavioral processes, such as the design of waste-separation policies [14], media [15], social interaction [16], community support, and investment in hardware facilities [17], and few have examined the results or focused on the effect of waste-separation behavior interventions on other PEBs.

Specific past or historical individual behaviors influence subsequent behaviors. In terms of environmental psychology, intervention strategies for one PEB may change the likelihood of individuals performing other PEBs, and different interventions have different shaping impacts [8,9,18,19,20]. If a certain type of intervention strategy can produce a positive impact and achieve a more effective result with less effort, governments will vigorously adopt and promote the strategy. By contrast, if intervention strategies lead to a reduction in engagements in other PEBs, policymakers can acknowledge and circumvent this limitation [9].

Conventionally, governments have used information interventions to spread the concept of waste separation and inspire and guide the public in its implementation. External information intervention plays an important role in persuading the public to engage in PEBs, and the difference between the appeal type (“what to say”) and message framing (“how to say it”) stimulates different individual motivations, reversing or further influencing subsequent behavioral choices [21,22]. Moreover, environmental appeals can motivate people to be altruistic and perform PEBs even if the benefits are minimal [8,18,23,24]. In comparison, monetary incentives stimulate egoistic motivation and encourage people to perform other PEBs only if they receive sufficient benefits [9,19,24,25]. In terms of sustainable development marketing, although loss-framed messages may be more persuasive because of increased social support [22,26], less attention has been paid to the impact of informational interventions on engagement in other PEBs. To fill this gap, in this study, we used an experimental method to explore the impacts of different waste-separation information interventions on public intention to perform other PEBs and their underlying processes.

Environmental risk perception is one of the prerequisites for the public to choose and implement environmental behaviors [27], and different types of message framing elicit emotional responses, thereby influencing risk perception [28]. Some studies have shown that the public’s perception of environmental risk for a particular environmental issue, such as air pollution, water pollution, and climate change, plays a significant role in promoting PEBs [14,15]. Furthermore, many scholars believe that the relationship between pro-environmental identity and the adoption of PEBs is well-established, because social identities are important predictors of behavior when a behavior is understood to be relevant to that identity [29,30]. Some studies have shown that information interventions that frame the benefits of a PEB in monetary versus moral terms can forge a new or stronger pro-environmental identity that is powerful enough to trigger the adoption of secondary behaviors, such as increased public preferences for environmentally friendly products, environmental intentions, and policy support [31,32]. Therefore, we examined the role of environmental risk perception and pro-environmental identity on the influence of the appeal type and message framing on residents’ intent to engage in PEBs.

Overall, our work is a pioneering study on the relationship between information interventions, perceived risk, pro-environmental identity, and environmental responsibility, and offers theoretical and applied insights. First, although numerous studies have examined appeal type or message framing on PEBs, such as green consumption [8,20,23], recycling [26,33], and energy savings [34], scholars have not paid sufficient attention to the interactive effect between appeal type and message framing. With this study, we are the first to explore the effectiveness of different types of appeals in motivating self-reported intent to perform a broad range of PEBs from the perspective of matching appeal type and message framing. Second, we investigated the mediating role of environmental risk perception and pro-environmental identity in this process and developed a comprehensive framework. Third, most earlier research focuses on cases from Western Europe, North America, and Australia, and has mostly focused on the effects of interventions in energy efficiency on other PEBs. This study aims to advance the literature by examining the effects of waste-separation information interventions on Chinese residents‘ intentions to perform other PEBs. Finally, our study provides important theoretical support for the selection and optimization of intervention strategies.

The remainder of this paper is structured as follows: In Section 2, we provide a review of the literature on PEBs, appeal type, and message framing, and develop our research hypotheses. Section 3, Section 4 and Section 5 present our research methodology and the results of Studies 1–3, which we used to investigate the impacts of appeal type and message framing, and the matching effect between them on intent to separate waste and engage in other PEBs. Section 6 and Section 7 present a discussion and our conclusions, respectively, describing theoretical and practical implications as well as the limitations of this study.

## 2. Literature Review and Hypotheses

### 2.1. Appeal Type and PEBs

Environmental appeals mainly refer to bringing awareness to the hazards caused by waste pollution and emphasizing to residents the contribution of waste separation to environmental protection, with the aim of fostering awareness of environmental norms and encouraging residents to voluntarily separate waste. Monetary incentives refer to the provision of financial rewards to residents who participate in waste separation or informing them of the fines they may face for not participating in waste separation to increase their motivation to separate [35].

PEBs are actions that are performed to minimize negative environmental impacts or ensure the sustainable use of natural resources, and include public PEBs, such as individual participation in environmental protection organizations and support or acceptance of public policies, and private PEBs, such as green consumption, classified recycling, and green travel [36]. The literature mainly explains individual PEBs and intentions, such as those based on the theory of planned behavior, value belief norms, awakening, and cognitive flexibility [37,38].

Regarding the factors influencing engagement in PEBs, interventions of individual-specific behaviors, such as encouragement, information, publicity, and policy design, can either positively or negatively affect both target and non-target PEBs [33]. Here, these positive impacts refer to an increase in one or more of the nontarget PEBs. For example, a call to save energy improves consumer willingness to engage in other environmental protection behaviors, and collecting taxes or fees on plastic bags improves consumer willingness to reuse [39]. Recycling garbage improves consumer willingness to buy environmentally friendly products [40]. Through a meta-analysis of interventions, Maki et al. (2019) found that interventions targeting intrinsic motivation and similarities in both target and nontarget PEBs were most likely to produce a positive impact on performing PEBs [5]. In this situation, the negative impacts indicate a reduction in engagement in other PEBs after performing the target PEB. For instance, consumers reduce their household water consumption after being exposed to water-saving information, whereas their power consumption strikingly increases [41]. Consumers who purchased electric cars exhibited lower environmental responsibility than those who bought traditional cars [42]. Engaging in plastic bottle recycling behaviors reduces consumer support for green funds [43].

Regarding the factors influencing PEBs, appeal type [19], individual behavioral characteristics [43], and behavioral attribution are the three most important [44]. While studying the effect of feedback on household electricity consumption, Fischer (2008) found that the most successful appeals combine the following features: they are frequently provided over a long period, they are presented in a clear and appealing manner, and they involve the use of computerized and interactive tools [34]. Evans et al. (2013) found that environmental appeals positively impact wastepaper recycling behavior, whereas monetary incentives or mixed appeals exhibit no such impact [9]. Steinhorst and Matthies (2016) also found that environmental appeals have a positive impact on other PEBs, whereas monetary incentives do not affect them [20]. Xu et al. (2018) examined the effects of two waste-separation interventions on engagement in other PEBs and found a positive pathway through changes in environmental concerns by environmental appeals, and a negative pathway through changes in pro-environmental identity by monetary incentives [24]. Similar results have been reported in field experiments [8,20]. The various types of appeal that emphasize environmental protection can stimulate individual altruistic behavior, whereas those emphasizing money rarely produce a positive impact on other PEBs.

Although a preliminary consensus has been reached on the impact of environmental appeals, this is not the case for its psychological mechanisms. From the perspective of social value theory, a few scholars state that environmental appeals can activate individual self-transcendental value perception and thus have a positive impact on engagement in other PEBs [6,45]. However, monetary incentives can activate the self-interest value perception of individuals, hindering such incentives from positively impacting other PEBs that do not result in self-interest. Based on self-cognition theory, certain researchers think that monetary incentives do not easily produce a positive impact on other PEBs because they weaken the intrinsic motivation or self-transcendental value that matches the environmental protection goal; thus, triggering the general engagement of individuals in other PEBs is difficult [7]. Therefore, we propose the following hypotheses:

**Hypothesis 1** **(H1):***Compared with monetary incentives, environmental appeals lead to a stronger intention to separate waste and other PEBs*.

### 2.2. Message Framing and PEBs

Message framing is a type of linguistic expression involving the sharing of information with similar meanings through different information-encoding methods, such as diction and tone, thus affecting the processing and decoding of the content by the audience, resulting in different cognitions and judgments [46]. Framing theory posits that the manner in which information is presented influences the individual responses. As a message design strategy, framing draws attention to certain aspects of information while simultaneously drawing attention away from other aspects. As a consequence of this presentational difference, individuals’ attitudes toward the topic become more aligned with the framed information, leading to more message-consistent effects, referred to as a framing effect [47]. Framing theory predicts that processing fluency will mediate the relationship between language frame and message-consistent outcomes, because the probability of drawing on framed beliefs through accessibility can be amplified by processing fluency or the ease with which something is processed [48]. In addition, when a message is judged as relevant, it is more likely to lead to message-consistent attitudes and behavior changes [49,50] because it is more likely to be remembered [51].

Gain-loss framing in environmental communication is a common equivalency frame that focuses on differences in how information is presented [47]. Gain-framed messages highlight the positive consequences of behaving in a certain manner. Conversely, loss-framed messages emphasize the risk of not behaving in a certain manner [52].

Van de Velde and Verbeke (2010) found that gain framing, compared with loss framing, can considerably improve the sensitivity of an individual to information and perceptual validity and strengthen consumer purchase intentions [46]. However, some scholars have found that loss-framed messages are more effective than gain-framed messages in motivating consumers to purchase healthcare products [53] or environmentally friendly biofuels [54]. Thus, the effectiveness of message frames depends on the level of cognitive elaboration, and these effects are moderated by the opportunity and motivation to process information [55].

Prospect theory [52] states that individuals widely differ in their sensitivity to gain and loss. When individuals have to decide between definite and potential acquisition, they tend to avoid risk and choose definite gains; in contrast, when deciding between determined and potential loss, they tend to pursue risk and choose potential loss. The sensitivities of individuals to gain and loss differ. Individuals are more sensitive to the pain of loss than to the pleasure of gain [56]. Compared with gain-framed messages, loss-framed messages stimulate negative emotions, which considerably increases pessimistic perceptions, helping to increase the acceptance and understanding of information [26,53]. In their study on prosocial behavior, Bullard and Penner (2017) found that using loss-framed messages to inform consumers of the negative consequences of not buying green products or engaging in charitable causes could stimulate their fear and lead to positive decisions [22]. PEBs are similar to other prosocial behaviors in terms of the psychological processes and behavioral responses to information processing. In this regard, the possible losses of environmental quality and personal health caused by the nonseparation of household waste are presented as potential losses to the public. People perceive that not separating waste will weaken their participation in environmental protection, and affect their own health and safety. Simultaneously, they will not receive short-term or long-term economic benefits. As the public tends to avoid losses or harm, this will lead to a positive response. Furthermore, if loss framing has a stronger persuasive effect on waste separation, residents may feel a stronger sense of social responsibility and regulation, performing more PEBs when loss framing is used. Thus, we constructed the following hypothesis:

**Hypothesis 2** **(H2):***Loss-framed messages are more likely than gain-framed messages to encourage the intention to separate waste and other PEBs*.

### 2.3. Appeal Type and Message Framing

During the processing of information, fluency and the persuasive effect can be improved by matching various factors affecting individual psychological processes [57]. Green and Peloza (2014) found that self- and other-benefit appeals that matched their private and public purchase settings, respectively, were more persuasive in influencing the willingness of consumers to purchase green products [58]. Chang et al. (2015) found that congruency between the loss frame and low-level construal, as well as the match between the gain frame and high-level construal, led to more positive outcomes in consumer attitudes and purchase intentions [59]. Segev et al. (2015) also found that loss/self-referencing messages produced more positive responses toward advertisements, brands, and purchase intentions than loss/environment-referencing messages [60]. They further explained that people pay more attention to messages that combine loss frames and self-referencing, because they naturally try to avoid negative impacts and risk. To alleviate the tension produced by the message, people adopt the behavior advocated in the message because they see it as a viable solution to the problem.

Loss framing matches with a firmer mindset [61], and negative appeals seem to be most effective when they specify the consequences of the behavior [62]. Therefore, people may struggle to deeply process information when matching environmental appeals with gain-framed messages, because the benefits of waste-separation behavior on the ecological environment are not immediately visible. Conversely, loss-framed messages can stimulate the sense of urgency in an individual and produce more positive associations and increased emotional processing [60].

When loss-framed messages are combined with monetary incentives, the messages express the possible loss caused by not participating in waste-separation behaviors. However, as this loss does not reduce their original wealth, individuals are not sensitive to the message. Contrastingly, when gain-framed messages are combined with monetary incentives, the messages express the financial benefits that individuals may receive by participating in waste sorting, which causes them to be sensitive to protecting both the environment and their self-benefit, thereby stimulating their positive emotions and responses. Prospect theory proposes that when individuals face a definite or potential gain, they tend to avoid risks and choose definite gain. Therefore, we inferred that monetary incentives aligned with gain-framed messages cause individuals to pay more attention to financial benefits. This definite gain can improve individual sensitivity to environmental protection and personal benefits; therefore, they engage in various other PEBs. Thus, we constructed the following hypotheses:

**Hypothesis 3a** **(H3a):***When exposed to environmental appeals, participants faced with loss-framing information will show stronger intention to separate waste and other PEBs than faced with gain-framing information*.

**Hypothesis 3b** **(H3b):***When exposed to monetary incentives, participants faced with gain-framing information will show stronger intention to separate waste and other PEBs than faced with loss framing information*.

### 2.4. The Mediating Role of Environmental Risk Perception

Stern (2000) classified the variables affecting environmental behavior into four main types: attitude, situational force, personal ability, and habit [4]. As an attitude factor, environmental risk perception reflects the cognition of the natural environment of an individual.

Individuals have stronger PEB intentions when they perceive the risk of their surroundings [63]. Akter and Bennett (2011) confirmed that heightened environmental risk perception is an important factor that drives the public to perform further actions to address environmental problems [64]. Zhao et al. (2011) found that scientific and environmental appeals helped improve individual perceptions of global-warming risk and indirectly and positively predicted individual support for emission-reduction policies, whereas political appeals did not play the same role [65]. Chang (2011) also found that environmental appeals indirectly and positively impacted environmental behavior through individual environmental value perception and intent to protect the environment [66]. Moreover, Meyers-Levy and Maheswaran (2004) found that both high-risk implications and high relevance foster systematic processing. When systematic processing dominates the thought behind risk perception, negatively framed messages have an advantage over positively framed messages, because of the former’s ability to predict intent to protect the environment [67].

Xu et al. (2018) found that increasing environmental concern is a mechanism underlying the positive impact of waste-separation environmental appeals; that is, environmental appeals enhance residents’ environmental concern, which then improves their level of participation in other PEBs [24]. No direct study has been conducted on the impact of monetary incentives on environmental risk perception. Evans (2013) found that loss-framed messages matched with monetary incentives easily influenced individual self-interest preferences and had no obvious short-term impact on engagement in PEBs [9]. Therefore, we propose that environmental risk perception may be a path through which appeal type and message framing influence the intent to separate waste and engage in other PEBs. Thus, we propose the following hypothesis:

**Hypothesis 4** **(H4):***Environmental risk perception plays a mediating role in the interaction between appeal type and message framing on the intention to separate waste and other PEBs*.

### 2.5. The Mediating Role of Pro-Environmental Identity

Pro-environmental identity refers to the extent to which individuals consider themselves environmentalists [68]. Pro-environmental identity not only directly and positively affects individual PEBs, but also plays a mediating role between implemented and subsequent PEBs [69] because self-identity is the core of self-conceptualization and can deeply influence individual behavior. Self-perception theory proposes that individuals act according to their views of themselves; otherwise, they experience cognitive dissonance. Therefore, the implemented PEBs can strengthen individual self-identity in relation to the environment, further promoting the engagement in other PEBs. Pro-environmental identity is strengthened when people participate in PEBs for environmental reasons, which may affect the performance of other PEBs. However, the absence of a strengthening effect when people engage in environmental protection for other reasons (e.g., money) is unlikely to impact other PEBs [70].

Van der Werff and Steg (2018) found that an emphasis on monetary benefits has an increased chance of weakening pro-environmental identity, whereas an emphasis on environmental benefits is more likely to enhance it [31]. In a quasi-experimental field study, Xu et al. (2018) also found that a monetary incentive strategy motivated individual self-interest, urged them to pay more attention to the pursuit of self-interested goals, and weakened the influence of pro-environmental identity on individual decision-making behavior [24]. This is an important reason why monetary strategy does not have a positive influence on pro-environmental identity and might even have a negative impact. As a typical altruistic behavior, the environmental behavior of residents must be driven by pro-environmental identity. However, monetary incentives may strengthen the external attribution of residents to their environmental protection behavior and generate the conviction that environmental protection does not come from environmentalism. When the level of risky implications and the relevance of information are low, heuristic processing dominates [67]. We inferred that when monetary incentives match loss framing, residents usually process information with marginal clues that can cause emotional resonance. This subjective and emotional state further improves the external attribution of environmental protection behavior.

Therefore, in this study, we assumed that a pro-environmental identity may be a path through which the process of appeal type and message framing has an influence on the intent to separate waste and other PEBs. In summary, we constructed the following hypothesis:

**Hypothesis 5** **(H5):***Pro-environmental identity plays a mediating role in the interaction between appeal type and message framing on the intention of waste separation and other PEBs*.

Notably, we found that environmental risk perception and pro-environmental identification were common factors that influenced engagement PEBs. However, through a literature review, we found a lack of studies exploring the relationship between the two. Therefore, in this study, we set these two factors as two parallel mediators.

We used three studies to test our hypotheses. Study 1 included a questionnaire and two focus-group interviews. Our aim was to identify the environmental appeal content for subsequent experiments and to improve the measurement scale for assessing the intent to perform other PEBs. Study 2 involved an experiment using two analysis of variance (ANOVA) tests to test H1, where environmental appeals led to a stronger intent to separate waste and engage in PEBs other than monetary incentives. Study 3 was a repeat of Study 2 to verify and test H1 and the main effect of message framing (H2). In addition, we used two-way factor ANOVA and a simple effect tests to explore the interaction between appeal type and message framing (H3a and H3b). We tested the mediating effect of pro-environmental identity and environmental risk perception using PROCESS 3.3 (H4 and H5). Figure 1 provides the study’s model.

## 3. Study 1

Study 1 involved the administration of a survey we developed to determine the appropriate environmental appeal types for subsequent experiments and to improve the scale used for evaluating the intent to engage in other PEBs.

### 3.1. Method

#### 3.1.1. Participants and Design

In May 2021, we distributed questionnaires randomly around a university in Hangzhou and 82 people filled it out. The final sample included 22 undergraduate students and 54 residents from the Shanxian community who completed the survey, with an average participant age of 30.66 (*SD* = 5.78); 36.84% were men. The educational level of 48.7% of the participants was a university or college degree, 30.2% had finished high school, and 21.1% had obtained less than a high school degree. Of the participants, 34.2% had a before-tax monthly personal income of less than RMB 5000; 53.4% from RMB 5000 to RMB 10,000; and 12.4% from RMB 10,000 or more. Upon completion of the study, the participants each received a gift of RMB 20.

#### 3.1.2. Procedure

We informed the participants that they would engage in a series of studies. The specific steps were as follows: First, we asked the participants to evaluate the four environmental motives for performing waste-separation behaviors on a seven-point scale. Our aim with the questionnaire was to identify environmentally appealing motivators for subsequent experiments. Second, we randomly selected and assigned 16 participants to two groups to conduct focus group interviews around the open theme of how they understood PEBs. Our aim with the focus group interviews was to improve the scale used for measuring the intent to perform of other PEBs.

#### 3.1.3. Measures

We measured the motives for participating in waste-separation behaviors using four items: saving resources and avoiding waste, protecting the environment and reducing pollution, getting motivated, and supporting government environmental protection work, on a seven-point scale (ranging from 1 = very unimportant to 7 = very important). We chose the item that received the highest score as the focus of environmental appeals in our subsequent experiments. Considering the potential disconnect between expert and public perceptions with respect to PEBs [71], we used the PEB conceptualizations of local respondents, based on the research of Lanzini and Thogersen (2014) and Xu et al. (2018), to guide the scale development in our study [24,72,73]. The understanding of the 16 respondents regarding PEBs provided an important reference in forming the index of the intent to perform other PEBs.

### 3.2. Results

According to the results, participants believed that environmental protection and pollution reduction were the most important factors affecting waste-separation behavior (*M* = 6.05, *SD* = 0.43). The scores for the other items were as follows: saving resources and avoiding waste (*M* = 5.50, *SD* = 0.83), motivation (*M* = 4.24, *SD* = 0.65), and supporting governmental environmental protection work (*M* = 4.11, *SD* = 0.79). Therefore, the environmental appeal focus for subsequent experiments was protecting the environment and reducing pollution.

The coding results of the focus group interviews showed that all respondents responded that waste separation, green travel, water and electricity savings, and green consumption constituted PEBs. For 10 of the 16 respondents, an extremely high participation cost, such as donating to environmental protection organizations, would reduce their intent to engage in PEBs. Most respondents (*N* = 9) expressed that their existing consumption habits and social trends, such as providing their own tableware when dining out and toiletries when traveling, affected their participation in PEBs. We referenced the measurement items used by Lanzini and Thogersen (2014) and Xu et al. (2018), using items such as green purchasing, green traveling, domestic electricity and water conservation, and supporting environmental protection policies [24,72]. Based on the coding results, we excluded items such as printing documents on both sides, and modified the willingness to engage in environmental citizenship actions to willingness to participate in various environmental protection activities. We added two items: not using disposable tableware when dining out and bringing your own toiletries when traveling. In addition, we divided recycling into three items (actively recycling and using idle items, bringing your own shopping bag when shopping, and reusing plastic bags) because these five PEBs are the things that many residents think they should be performing, but adhering to these activities is a challenge. Consequently, we applied 11 items to measure the intent to engage in other PEBs, including the degree of participation in private PEBs, such as green travel, water and electricity savings, environmental policy support, and civic behavior. Subsequently, we submitted the scale for examination by three scholars in the field of environmental protection who agreed with the items.

## 4. Study 2

We developed Study 2 to test the appeal types. We predicted that individuals who receive environmental appeals would have a stronger intent to separate waste and perform other PEBs than those who receive monetary incentives (H1).

### 4.1. Method

#### 4.1.1. Participants and Design

We recruited 133 adults currently residing in China, using a paid sample service from a research website (www.wjx.cn, (accessed on 30 June 2021)). The final sample comprised 126 participants, of which seven failed to pass the quality control check because of their negative attitudes. Their average age was 24.25 years (*SD* = 4.45), and 31.74% of the participants were men. The educational level of 50.8% of the participants was a university or college degree, 35.7% had finished high school, and 13.5% obtained less than a high school degree. The before-tax monthly income of 33.3% of the participants was less than RMB 5000; 56.3% from RMB 5000 to RMB 10,000; and 10.4% from RMB 10,000 or more. None of the respondents had participated in a similar study. We randomly assigned participants to one of three conditions (environmental appeal, monetary incentive, or control).

#### 4.1.2. Procedure and Manipulations

The experiment comprised three phases. In the first phase, we asked the environmental appeal (*N* = 42) and monetary incentive (*N* = 42) groups to view relevant information material on waste separation. The environmental appeal group viewed material such as “Proper waste separation can prevent harmful substances from entering water sources….” The monetary incentive group viewed the material such as “If you correctly separate waste, you can accumulate points in exchange for eggs, soy sauce….” We asked the control group (*N* = 42) to spend one minute performing a number of addition and subtraction exercises. Appendix B provides detailed information on the experimental content. Next, to enhance their involvement, we asked the participants in the three groups to provide a relevant response after reading the provided intervention information as follows: the environmental appeal group was tasked with, “June 5 is world environment day. To increase people’s attention to environmental issues, the community has launched publicity activities for environmental protection. Please write a slogan for the activity;” the monetary incentive group was tasked with, “February 27 is the third anniversary of the promulgation of ‘The Implementation Plan of Household Waste Separation.’ To promote waste separation, the community plans to increase material rewards for residents. Please provide suggestions and help the community further improve the relevant monetary incentive system.” This item was not analyzed in depth. The control group was asked to spend three minutes performing a number of addition and subtraction exercises.

In the second phase, the environmental appeal and monetary incentive groups completed a questionnaire that we designed to assess the effects of the manipulation based on the appeal type. These questions, such as “To what extent do you think the above information describes the perspective of environmental appeal/monetary incentive?” were answered on a seven-point scale (ranging from 1 = very low to 7 = very high). The control group was asked to spend 1 min performing a number of addition and subtraction exercises.

In the third phase, participants responded to a questionnaire related to the dependent variables and demographic questions, such as gender, age, monthly income, and education level, which were controlled for in the subsequent analysis. The dependent variables included intent to separate waste and perform other PEBs. We measured the intent to separate waste (α = 0.87) using five items on a seven-point scale (ranging from 1 = strongly disagree to 7 = strongly agree), which we adapted from Kaiser (2003) [74]. Sample items included “I will separately store the classified garbage before putting it in” and “When handling garbage, I will put it into different categories.” The five items reported in Appendix A were averaged to form an index of intention of waste separation.

PEBs can be measured by directly observing the behavior of interest in the field or laboratory. Alternatively, researchers can rely on the retrospective reports of participants of their own behavior [37]. The results of a meta-analysis by Kormos and Gifford (2014) revealed a medium to large effect size (r = 0.46) for the correlation between self-reported and objectively assessed PEBs [75]. Self-reported measures have also been used to assess individual propensity to behave pro-environmentally in different situations [76,77]. Whether the propensity measured using multi-item PEB scales is unidimensional or multidimensional has been the subject of debate [78,79] and is dependent on the individual operational definition of unidimensionality [37]. For example, Kaiser and Wilson (2004) found that a six-dimensional model more accurately fit their data than a unidimensional model, and self-reports of different PEBs appeared to be sufficiently related to be combined into a meaningful domain-general index [78]. In this study, we measured the intent of residents to engage in other PEBs (*α* = 0.95) through self-reporting using a unidimensional scale. The PEBs we used in Study 2 were the same as those used in Study 1. The results of exploratory factor analysis identified one common factor (KMO test for sampling adequacy = 0.87), and the factor loadings ranged from 0.55 to 0.70. Participants responded to items such as “Are you willing to X (other PEBs)?” on a seven-point scale (ranging from 1 = never to 7 = always). We averaged the 11 items reported in Appendix A to form an index of the intent to engage in other PEBs.

### 4.2. Results

#### 4.2.1. Manipulation Checks

The independent sample *t*-test results revealed that in the environmental appeal group, participants tended to place a higher value on environmental appeals (*M* = 5.55, *SD* = 1.06) than monetary incentives (*M* = 4.10, *SD* = 0.62), *t* (82) = 7.65, *p* < 0.001, Cohen’s *d* = 1.67. In the monetary incentive group, the monetary incentive scores (*M* = 5.67, *SD* = 0.72) were significantly higher than those for environmental appeals (*M* = 4.38, *SD* = 0.76), *t* (82) = 7.93, *p* < 0.001, Cohen’s *d* = 1.74. Thus, we concluded that the appeal-type manipulation was successful.

#### 4.2.2. Hypothesis Testing

The results of the two ANOVA tests showed that intent to separate waste varied according to appeal type (*F* (2, 123) = 43.01, *p* < 0.001, *η*^2^ = 0.41). The results of a further post-test (LSD) revealed that the intent of those who received environmental appeals (*M* = 5.85, *SD* = 0.75) was significantly stronger than that of those who accepted monetary incentive appeals (*M* = 4.46, *SD* = 0.99; 95% CI: (1.03, 1.73); *p* < 0.001, Cohen’s *d* = 1.58) and those who did not receive any appeals (*M* = 4.41, *SD* = 0.98; 95% CI: (1.09, 1.79), *p* < 0.001, Cohen’s *d* = 1.65). We found no significant difference between the monetary incentive and control groups (*p* = 0.745).

The intent to engage in other PEBs varied according to appeal type *(F* (2, 123) = 58.28, *p* < 0.001, *η*^2^ = 0.49). The results of a further post-test (LSD) revealed that the intent of those who received environmental appeals (*M* = 5.57, *SD* = 1.14) was significantly stronger than that of those who received monetary incentive appeals (*M* = 4.06, *SD* = 0.53, 95% CI: (1.18, 1.83); *p* < 0.001, Cohen’s *d* = 1.70) and those who did not receive any appeal (*M* = 4.01, *SD* = 0.34; 95% CI: (1.24, 1.88); *p* < 0.001, Cohen’s *d* = 1.85). We found no significant differences between the monetary incentive and control groups (*p* = 0.740).

H1 was supported by the data; that is, environmental appeals led to a stronger intent to separate waste and perform other PEBs (Figure 2).

## 5. Study 3

We designed Study 3 to directly test message framing, examine the interaction between appeal type and message framing, and explore the intermediary effect of pro-environmental identity and environmental risk perception.

### 5.1. Method

#### 5.1.1. Participants and Design

We recruited 335 adults from Wenxin and Jinjiang communities in Hangzhou, China. The participants in the three studies were completely independent. The final sample comprised 320 participants, 15 of whom failed to the pass the quality control check because of their negative attitudes. The average age was 28.28 years (*SD* = 6.13), and 42.80% were men. The educational level of 34.4% of the participants was a university or college degree, 47.2% had finished high school, and 18.4% obtained less than a high school degree. Of the participants, 28.7% had a before-tax monthly personal income of less than RMB 5000; 55.3% from RMB 5000 to RMB 10,000; and 16% from RMB 10,000 or higher. We randomly and equally assigned participants to one of the four conditional groups in a 2 (appeal type: environmental vs. monetary) × 2 (message framing: gain vs. loss) between-subjects design (*N* = 80, 80, 80, and 80). Upon completion of the experiment, all the participants received a gift of RMB 20.

#### 5.1.2. Procedure and Manipulations

We informed the participants that they would take part in a series of experiments. The specific steps were as follows: First, referring to the experimental design methods used by Steinhorst and Matthies (2016) and White et al. (2011) [20,61], the first part of the situational stimulus material used gain- or loss-framed messages to describe the results of waste-separation behaviors owing to monetary or environmental factors, such as “Through the correct separation of household waste…”; “If household waste is not properly separated…”; “Of household waste, 30–40% can be recycled…”; and “Of domestic waste, 30–40% can be recycled….” Detailed information is presented in Appendix B. Next, to enhance their involvement, we asked participants in each of the two gain-framing groups to write down the environmental benefits and monetary benefits they would receive from separating waste. Participants in each of the two loss-framing groups were asked to write down the environmental losses or monetary losses they might experience from not participating in waste separation. This item was not analyzed in depth. Second, the participants responded to the designed manipulation-check questions, such as “To what extent do you think the information describes the benefits of waste sorting?” and “To what extent do you think the information describes the losses that would result from not waste sorting?” on a seven-point scale (ranging from 1 = very low to 7 = very high). The manipulation-check questions for appeal type were similar to those in Study 2. We asked all participants to answer all four manipulation-check questions. Finally, we asked the participants to complete a series of self-administered questionnaires, including questions related to the mediators, dependent variables, and demographic questions such as gender, age, monthly income, and education level, which were controlled for in the subsequent analysis.

We measured pro-environmental identity using six items (α = 0.79) on a seven-point scale (ranging from 1 = strongly disagree to 7 = strongly agree), which we adapted from Whitmarsh and O’Neill (2010) [29]. The items included statements such as “I consider myself an environmental consumer” and “I think I am a person who is very concerned about environmental issues”. We averaged the six items reported in Appendix A to form an index of pro-environmental identity. We measured environmental risk perception using a twelve-item scale (*α* = 0.90), which we had adapted from the 2013 China Comprehensive Social Survey. Participants rated the degree to which the items reflected their attitudes on a seven-point scale (ranging from 1 = strongly disagree to 7 = strongly agree). The items included questions such as “Do you agree that there is currently a risk of air pollution in China?” and “Do you agree that there is currently a risk of water pollution in China?” We averaged the 12 items reported in Appendix A to form an environmental risk-perception index. The scales we used to measure intent to separate waste (α = 0.82) and other PEBs (α = 0.84) were the same as those used in Study 2.

### 5.2. Results

#### 5.2.1. Manipulation Checks

The results of the independent sample *t*-test revealed that in the loss-framing condition group, the participants tended to place a higher value on loss (*M* = 5.85, *SD* = 1.15) than on gain framing (*M* = 3.61, *SD* = 1.70), *t* (318) = −13.78, *p* < 0.001, Cohen’s *d* = 1.54. Gain scores (*M* = 5.51, *SD* = 1.27) were significantly higher in the gain-framing group than in the loss-framing group (*M* = 3.96, *SD* = 1.88), *t* (318) = 8.66, *p* < 0.001, Cohen’s *d* = 0.97. Thus, we conclude that the message-framing manipulation was successful.

The results of the independent sample *t*-test revealed that in the environmental condition group, the participant scores for environmental appeals (*M* = 5.94, *SD* = 1.14) were significantly higher than those for monetary incentives (*M* = 4.03, *SD* = 1.81), *t* (318) = 11.25, *p* < 0.001, Cohen’s *d* = 1.26. In the monetary condition group, the monetary incentive scores of the participants (*M* = 5.38, *SD* = 1.55) were significantly higher than those for environmental appeals (*M* = 2.64, *SD* = 1.54); *t* (318) = −15.88, *p* < 0.001, Cohen’s *d* = 1.77. Therefore, we considered that the appeal-type manipulation was successful.

#### 5.2.2. Hypothesis Testing

We performed an independent sample *t*-test to test the strengths H1 and H2. As predicted, we found a significant main effect of appeals. The intent to separate waste in the environmental appeal group (*M* = 5.43, *SD* = 0.78) was significantly stronger than in the monetary incentive group (*M* = 4.74, *SD* = 0.60); *t* (318) = 8.82, *p* <0.001, Cohen’s *d* = 0.99. The test results for the intent to perform other PEBs were similar (*M*_environmental_ = 5.40, *SD* = 0.76; *M*_monetary_ = 4.94, *SD* = 0.50; *t* (318) = 6.33, *p* < 0.001, Cohen’s *d* = 0.72), which again supported H1. However, the main effects of message framing were insignificant (*p* > 0.05); therefore, H2 was not supported.

In the test of H3a and H3b, the results of two-way factorial ANOVA tests indicated that for intent to separate waste, the main effect of appeal type was significant (*F* (1, 316) = 83.48, *p* < 0.001, *η*^2^ = 0.21). The environmental appeal group scored higher than the monetary incentive group (*M*_environmental_ = 5.43, *SD* = 0.78; *M*_monetary_ = 4.74, *SD* = 0.60, Cohen’s *d* = 0.99), and we found an interactive effect between appeal type and message framing (*F* (1, 316) = 24.25, *p* < 0.001, *η*^2^ = 0.07). For intent to engage in other PEBs, the main effect of appeal type was significant (*F* (1, 316) = 39.05, *p* < 0.001, *η*^2^ = 0.11). The environmental appeal group scored higher than the monetary incentive group (*M*_environmental_ = 5.38, *SD* = 0.80; *M*_monetary_ = 4.92, *SD* = 0.51; Cohen’s *d* = 0.69). We found an interactive effect between appeal type and message framing (*F* (1, 316) = 28.02, *p* < 0.001, *η*^2^ = 0.08). Table 1 provides additional details.

To further understand the interaction between the appeal type and message framing, we performed simple effect tests.

Intent to separate waste. The results suggested that intergroup differences were significant. Consistent with our arguments, environmental appeals elicited a stronger intent to separate waste when matched with loss framing (*M* = 5.65, *SD* = 0.84) than with gain framing (*M* = 5.21, *SD* = 0.66), *F* (1, 316) = 16.84, *p* < 0.001, *η*^2^ = 0.05). Similarly, monetary incentives, when matched with gain framing (*M* = 4.90, *SD* = 0.61), generated a stronger intent to separate waste than when matched with loss framing (*M* = 4.59, *SD* = 0.56), *F* (1, 316) = 8.19, *p* = 0.005, *η*^2^ = 0.03. In addition, the simple effect of appeal type was significant in both the gain-framing condition group (*F* (1, 316) = 8.97, *p* = 0.003, *η*^2^ = 0.03) and the loss-framing condition group (*F* (1, 316) = 98.86, *p* < 0.001, *η*^2^ = 0.24). The environmental appeal group scored higher than the monetary incentive group (*M*_environmental-gain_ = 5.21, *SD* = 0.66; *M*_monetary-gain_ = 4.90, *SD* = 0.61, Cohen’s *d* = 0.49; *M*_environmental-loss_ = 5.65, *SD* = 0.84; *M*_monetary-loss_ = 4.59, *SD* = 0.59, Cohen’s *d* = 1.46), as illustrated in Figure 3A. However, the main effect of message framing was not statistically significant (*p* = 0.380).

Intent to Engage in Other PEBs. The results suggested significant intergroup differences. Consistent with our arguments, environmental appeal, when matched with loss framing (*M* = 5.60, *SD* = 0.83) was a stronger inducer of intent to perform other PEBs than when matched with gain framing (*M* = 5.16, *SD* = 0.71), *F* (1, 316) = 18.40, *p* < 0.001, *η*^2^ = 0.06. Similarly, monetary incentives, when matched with gain framing (*M* = 5.09, *SD* = 0.50), provided a stronger incentive for engaging in other PEBs than when matched with loss framing (*M* = 4.77, *SD* = 0.46), *F* (1, 316) = 10.22, *p* = 0.002, *η*^2^ = 0.03 Additionally, in the loss-framing condition group, the simple effect of appeal type was significant (*F* (1, 316) = 66.61, *p* < 0.001, *η*^2^ = 0.17). The environmental appeal group scored higher than the monetary incentive group (*M*_environmental-loss_ = 5.60, *SD* = 0.83; *M*_monetary-loss_ = 4.77, *SD* = 0.46, Cohen’s *d* = 1.24); however, this was not significant in the gain-framing condition group (*F* (1, 316) = 0.46, *p* = 0.50, *η*^2^ = 0.00; Figure 3B). The main effect of message framing was not statistically significant (*p* = 0.440). Thus, we confirmed our interpretation of the interactive effects between appeal type and message framing.

To test H4 and H5, we first conducted two-way factorial ANOVAs on the two mediator variables to examine the impact of the four conditions on environmental risk perception and pro-environmental identity.

Environmental Risk Perception. The main effect was significant for appeal type (*F* (1, 316) = 35.94, *p* < 0.001, *η*^2^ = 0.1). The environmental appeal group scored higher than the monetary incentive group (*M*_environmental_ = 5.40, *SD* = 0.62; *M*_monetary_ = 4.88, *SD* = 0.90, Cohen’s *d* = 0.67), but not for message framing (*p* = 0.309). The interaction between appeal type and message framing was also significant (*F* (1, 316) = 6.53, *p* = 0.011, *η*^2^ = 0.02). The results of further simple effect analysis revealed that the simple effect of appeal type was significant for both gain framing (*F* (1, 316) = 5.91, *p* = 0.016, *η*^2^ = 0.02) and loss framing (*F* (1, 316) = 36.56, *p* < 0.001, *η*^2^ = 0.10). The environmental appeal group scored higher than the monetary incentive group (*M*_environmental-gain_ = 5.25, *SD* = 0.70; *M*_monetary-gain_ = 4.95, *SD* = 0.95, Cohen’s *d* = 0.36; *M*_environmental-loss_ = 5.55, *SD* = 0.50; *M*_monetary-loss_ = 4.82, *SD* = 0.86, Cohen’s *d* = 1.04).

Pro-environmental Identity. First, the main effects of both appeal type (*F* (1, 316) = 3.94, *p* = 0.048, *η*^2^ = 0.01) and message framing (*F* (1, 316) = 5.66, *p* = 0.018, *η*^2^ = 0.02) were significant. The environmental appeal group scored higher than the monetary incentive group (*M*_environmental_ = 5.15, *SD* = 0.85; *M*
_monetary_ = 4.96, *SD* = 0.94, Cohen’s *d* = 0.21), and the gain-framing group scored higher than the loss-framing group (*M*_gain_ = 5.17, *SD* = 0.88; *M*_loss_ = 4.94, *SD* = 0.91, Cohen’s *d* = 0.26). The interaction between appeal type and message framing was also significant (*F* (1, 316) = 7.86, *p* = 0.005, *η*^2^ = 0.02). The results of further simple effect analysis revealed that although the simple effect of appeal type was insignificant in the gain-framing condition group (*p* = 0.563), it was significant in the loss-framing condition. We found that environmental appeals (*M* = 5.18, *SD* = 0.74) were stronger influencers of pro-environmental identity than monetary incentives (*M* = 4.70, *SD* = 0.99), Cohen’s *d* = 0.55, *F* (1, 316) = 11.47, *p* = 0.001, *η*^2^ = 0.04, Figure 4).

Second, following the moderated mediation-effect-test method proposed by Hayes (2013) [80], we ran two separated moderated mediation models and introduced message framing as the independent variable, appeal type as the moderator, environmental risk perception and pro-environmental identity as the mediators, and intent to separate waste and perform other PEBs as dependent variables using bias-corrected asymmetric bootstrapping of the 95% CI based on 5000 bootstrap samples (Model 7 of PROCESS 3.3).

Table 2 presents the results of the regression analysis of the mediating effects. As shown in the table, the interaction between message framing and appeal type significantly predicted environmental risk perception *(B* = −0.44, SE_B_ = 0.17, *p* < 0.05). We found a positive conditional effect of message framing on environmental risk perception through environmental appeal (coefficient = 0.31, SE = 0.12, *p* = 0.012, CI (0.0681, 0.5471)), and a nonsignificant effect for monetary incentives (coefficient = −0.13, SE = 0.12, *p* = 0.278, CI (−0.3719, 0.1071)). The interaction between appeal type and message framing also significantly predicted pro-environmental identity (*B* = −0.55, SE_B_ = 0.20, *p* < 0.01). We found that the conditional effect of message framing on pro-environmental identity was negative for monetary incentives (coefficient = −0. 51, SE = 0.14, *p* < 0.001, CI (−0.7838, −0.2362)) and nonsignificant for environmental appeals (coefficient = 0.04, SE = 0.14, *p* = 0.763, CI (−0.0.2319, 0.3157)). Environmental risk perception positively affected the performance of waste separation (*B* = 0.37, SE_B_ = 0.05, *p* < 0.001) and other PEBs (*B* = 0.16, SE_B_ = 0.05, *p* < 0.01). Environmental identity also significantly affected engagement in waste separation (*B* = 0.29, SE_B_ = 0.05, *p* < 0.001) and other PEBs (*B* = 0.12, SE_B_ = 0.04, *p* < 0.01).

Third, following Hayes (2014) [81], we calculated the index of moderated mediation to examine the differences between the conditional indirect effects. An index significantly different from zero indicates that the conditional indirect effects differ.

In support of H4 and H5, the indices of moderated mediation were significantly different from zero for dependent variables. This indicated that the indirect effect of message framing through environmental risk perception on intent to separate waste significantly differed by appeal type (index = −0.16, CI (−0.3138, −0.0313)). Specifically, in the environmental appeal condition group, the indirect effect of message framing on intent to separate waste was positive (coefficient = 0.11, SE = 0.04, CI (0.0416, 0.1975)), and the perception risk of the environment of the match between environmental appeals and loss framing was significantly higher than that between environmental appeals and gain framing, i.e., the environmental risk perception of the participants was further stimulated when environmental appeals matched with loss framing, thus generating a positive impact on waste separation. However, in the monetary incentive condition group, the indirect effects of message framing on intent to separate waste were not significant (index = −0.05, CI [−0.1617, 0.0551]). This indicated that the indirect effect of message framing through environmental risk perception on the intent to engage in other PEBs significantly differed by appeal type (index = −0.07, CI (−0.1707, −0.0103)). Specifically, in the environmental appeal condition group, the indirect effect of message framing on the intent to engage in other PEBs was positive (coefficient = 0.05, SE = 0.02, CI (0.0125, 0.1042)). The environmental risk perception of environmental-loss messages was significantly higher than that of environmental-gain messages, i.e., the environmental risk perception of the participants was further stimulated when environmental appeals matched with loss framing, thus generating a positive impact on engagement in other PEBs. However, in the monetary incentive condition group, the indirect effect of message framing on the intent to perform other PEBs was not significant (index = −0.02, CI [−0.0860, 0.0205]).

The findings also indicated a significant difference by appeal type in the indirect effect of message framing through pro-environmental identity on the intent to separate waste (index = −0.16, CI (−0.3218, −0.0427)). Specifically, in the monetary incentive condition group, the indirect effect of message framing on intent to separate waste was negative (coefficient = −0.15, SE = 0.05, CI (−0.2609, −0.0557)). The pro-environmental identity of the match between monetary incentives and loss framing was significantly lower than that between monetary incentives and gain framing. In other words, when monetary incentives matched loss framing, the sense of pro-environmental identity of the participants weakened, thus reducing the possibility of a positive impact on waste-separation behaviors. However, in the environmental appeal condition group, the indirect effect of message framing on intent to separate waste was not significant (index = 0.01, CI [−0.0585, 0.1040]).

The indirect effect of message framing through pro-environmental identity on the intent to perform other PEBs significantly differed by appeal type (index = −0.07, CI (−0.1718, −0.0070)). Specifically, in the monetary incentive condition group, the indirect effect of message framing on the intent to engage in other PEBs was negative (coefficient = −0.06, SE = 0.03, CI (−0.1431, −0.0096)), and the pro-environmental identity of monetary-loss messages was significantly lower than that of monetary-gain messages, i.e., when monetary incentives matched loss-framing, the sense of pro-environmental identity of the participant weakened, thus reducing the possibility of a positive impact on engagement in other PEBs. However, in the environmental appeal condition group, the indirect effect of message framing on the intent to perform other PEBs was not significant (index = 0.01, CI [−0.0243, 0.0518]). Thus, H4 and H5 was supported. Table 3 provides further details.

## 6. Discussion

In this study, we explored the process through which environmental appeals and monetary incentives influence the intent to separate waste and to engage in other PEBs by conducting a survey and two experiments. From the results of Study 1 (survey), we determined that the main experiment should focus on protecting the environment and reducing pollution when designing environmental appeals and improving the test scales to determine the intent to perform other PEBs. The results of Study 2 revealed that the positive effect of environmental appeals was stronger than that of monetary incentives, and we found little evidence that monetary incentives were effective. Based on the results of Study 2, we further investigated the effect of message framing and its interactive effect on the intent to separate waste and engage in other PEBs in Study 3. The results indicated that although we found no significant difference in the influence of different message framings on the intent to separate waste or to engage in other PEBs, we identified an interactive effect between appeal type and message framing. In the case of environmental appeals, the impact of loss framing on the intent to separate waste and perform other PEBs was stronger than that of gain framing. The use of monetary incentives produced a stronger impact with gain framing on the intent to separate waste and engage in other PEBs.

Further analysis of the moderated mediation effect indicated that the indirect effect of message framing on the intent to separate waste and perform other PEBs through environmental risk perception or pro-environmental identity was moderated by appeal type. Specifically, environmental appeals positively regulated the impact of message framing on environmental risk perception. When the environmental appeals matched the loss framing, the perception of environmental risk substantially improved, thus promoting the intent to separate waste and engage in other PEBs. In addition, monetary incentives negatively regulated the impact of message framing on pro-environmental identity. When monetary incentives were matched with loss framing, pro-environmental identity weakened, thereby reducing the positive impact on waste separation and other PEBs.

In this study, we used environmental appeals and monetary incentives to test individual intent to separate waste and engage in other PEBs. We found that the positive effect of environmental appeals was stronger than that of monetary incentives. This conclusion is consistent with those of previous studies [8,9,20]. However, some researchers found no significant difference between the effects of environmental appeals and monetary incentives on waste-separation behavior [24]. The intention–implementation gap can manifest in different ways in the formation of PEBs, which may be an important reason why the effects on behavioral intentions were stronger than those on behavior [82].

Findings on framing research in environmental contexts have been inconsistent. Some researchers have found a stronger impact of loss-framing on the consumption of green goods or philanthropy than for gain-framing. However, recent research has found that the effects of message framing on food-waste-prevention intentions [83] and reducing air conditioning use in university dormitories [84] are not significant. Early research also found that positive frames produced more favorable attitudes toward curbside recycling [85,86] and water conservation [86]. In addition, appeals containing both positively and negatively framed statements have been shown to increase recycling behavior [87]. In this study, we found that, although message framing had an influence, the influence differed depending on the appeal type. This difference may have been related to the research context. Although green product consumption and charitable activities are also PEBs, the range of PEBs in this study that we measured was broader. Framing effects occur through processing fluency and applicability, both of which lead to greater frame-consistent perceptions [50]. In addition, framing literature indicates that the involvement level of the message recipient and the interaction of the depth of processing are important factors that moderate the persuasiveness of message frames [62]. Among the previous studies that found loss framing to be more effective for PEBs, most experimental materials were related to specific green advertising information. High participant involvement can rapidly form certain advertising attitudes and purchase intentions that may influence message framing. In contrast, because of the vigorous promotion of waste classification by the Chinese government, participants were familiar with the experimental materials, which means that the fluency and applicability of the information were at normal levels. However, participants were not highly engaged in this topic (although we implemented a few measures to improve involvement). Topic involvement refers to the importance and relevance of how an individual perceives a topic, which positively affects their motivation to process information [88]. Therefore, we observed no significant differences in the main effects of message framing. This result is consistent with that of a previous study, which reported that the loss framework was effective only if it combined specific information and situational factors of participation [61]. Factors such as issue salience (low vs. high) [86], recommended behavior (conservation/taking less vs. recycling/doing more), and the beneficiary of the behavior’s effects (current vs. future generations) [89] affect the persuasive effect of framing. Therefore, there is still room for further research on message framing, and there is an intrinsic relationship between framing design and research context that needs to be explored.

The interactive effect between appeal type and message framing may generally be driven by the perception of fluency and ease of information processing [40,61]. These findings support our results to a certain extent. Matching environmental appeals with loss framing can convey a sense of urgency to individuals and encourage them to engage more in environmental protection actions. Matching monetary incentives and gain framing is an approach to stimulating the enthusiasm to engage in individual environmental protection behaviors, leading to a stronger impact on the intent to separate waste and perform other PEBs.

In this study, we found that the message-framing process that indirectly affected the intent to separate waste and engagement in other PEBs through environmental risk perception was moderated by appeal type, and its indirect impact was positive for environmental appeal but not significant for monetary incentives. Furthermore, the environmental risk perception of the participants was further stimulated when environmental appeals matched the loss framing, thus generating a positive impact on waste separation and other PEBs. This result is similar to previously reported findings [24,66], that is, environmental appeals positively impact environmental behaviors through environmental attitudes or concerns. In addition, the message-framing process that indirectly affects the intent to separate waste and perform other PEBs through a pro-environmental identity was moderated by the appeal type. Its indirect impact was negative for monetary incentives and insignificant for environmental appeal. Furthermore, when monetary incentives matched the loss-framing, the pro-environmental identity of participants weakened, which thus reduced the possibility of a positive impact on engagement in waste separation and other PEBs.

Researchers have frequently described pro-environmental identity as having a positive impact on PEBs, and the relationship between environmental identity and the adoption of green behaviors is well-established [29,30]. For example, Werff (2014) found that reminding people of previous PEBs strengthened their pro-environmental identity and promoted engagement in other PEBs [69]. This finding is consistent with ours. Financial incentives may even have a negative impact because targeting PEBs with extrinsic motivators may hinder (i.e., “crowd out”) the intrinsic motivation to engage in other PEBs [18,90,91]. Xu et al. (2018) found that the crowding-out effect of external rewards on pro-environmental identity inhibited the positive impact on engagement in other PEBs and increased the probability of a negative impact, because monetary inducement can weaken individual intrinsic motivation to engage in altruistic behavior [24]. Using a meta-analysis of interventions, Maki et al. (2019) confirmed that for the intent to engage in other PEBs, incentive interventions reduced the positive impact relative to non-incentive interventions [5]. On this basis, we further found that monetary loss messages actually decreased environmental identity, so the likelihood of separating waste and performing other PEBs decreased.

For all the PEBs we investigated in this study, the index of moderated mediation was small, fluctuating around −0.10. This finding may have been related to the behavioral intention test method. Carrico (2021) proposed that the effects of behavioral interventions vary in direction and magnitude as a function of how the behaviors are measured [92]. With improvements in the experimental protocol, the crowding-out effect of intrinsic motivation and improvement in environmental risk perception may be enhanced with continuous information intervention, leading to a more remarkable moderation effect. In addition, our aim in this study was to explore the influence process of appeal type and message framing on the intent of residents to engage in PEBs. We did not investigate the specific cognition of residents in terms of various behaviors in detail, so we could not accurately judge the existence of differences between various PEBs. Although Kaiser and Wilson (2004) found that self-reports of different PEBs were sufficiently related to be combined into a meaningful domain general index [78], some scholars have found systematic differences between the propensities to engage in different PEBs [6,24,72]. Therefore, this problem should be addressed through more detailed studies in the future.

## 7. Conclusions

Information intervention is a basic method used to persuade individuals to engage in PEBs. In this study, we adopted an experimental method to explore ways to improve the positive impact of information interventions on the intent of residents to engage in waste separation and other PEB.

Our findings revealed a few strategic methods for using appropriate messaging strategies to persuade people to increase their intent to separate waste and engage in other PEBs. On the basis of these findings, we provide points of reference for government departments, communities, and environmental protection agencies in terms of waste separation and other areas of application. First, the government should focus on accurately identifying the effect of information interventions and then incorporating the intervention into the framework of policy design and evaluation to fully leverage the possible benefits. For implementation, rather than pursuing monetary incentives, publicity, education, and the cultivation of the public recognition of environmental protection should be emphasized. Specifically, the loss framework can be adopted for the general public to increase environmental risk perception from the perspectives of environmental protection and resource conservation, thus encouraging the public to participate in waste sorting. Second, for groups with a desire for money and communities with low average income, a gain framework with monetary incentives should be adopted to increase enthusiasm for participating in garbage classification. Careful design and collocation of information will increase public environmental risk perception, and more explicit strategies, such as positive feedback or labeling them as “environmentalists” would be the more effective choices to further strengthen individual environmental identity. This can positively impact their intent to engage in other PEBs and indirectly promote the development of environmental protection causes and sustainable consumption behaviors. Finally, as the overall level of ecological behavior norms is still low and internalization of biospheric values is generally lacking for the Chinese public [35], the government needs to be cautious when using monetary loss information because a crowding-out effect with monetary loss information may occur.

Although our findings provide a reference for PEB research, our study had certain limitations. First, a correlation exists between the intent to separate waste and engagement in other PEBs. PEBs have spillover effects [33,39,40,41,43]. However, in this study, we ran two separate moderated mediation models: one predicting the intent to separate waste and another predicting the intent to perform other PEBs. We did not explicitly consider the relationship between the two; however, the hidden relationship between them may have affected the study results. Second, we considered the effect of information intervention only in the short term and did not address long-term trends. Finally, the information intervention measures and identification-mediation process we described in this study explain the creation of a few impacts. Future research can explore a more in-depth analysis of the psychological mechanism behind these impacts and focus on the possible paths of impact among appeal types, message framing, and other mediators. Moreover, because observational data on individual PEBs are difficult to obtain, we relied on the literature and used a self-report method to measure public tendency and intent to engage in waste separation and other PEBs. However, to a certain extent, the type of behavioral measurement and research settings altered the associations between identity and PEBs [93]. Therefore, the methodological limitations may have affected the external validity of our results. In addition, the error caused by the potential social expectation effect of participants is difficult to avoid. Finally, we explored the factors that affected the intent of residents to engage in PEBs. However, important differences exist in ethical behavior across different cultures, and the intent–implementation gap is a particularly salient feature in promoting individual PEBs in the context of Chinese culture [82]. Therefore, in future comparisons of PEBs, researchers should consider cross-cultural differences, as well as directly measuring the PEBs of participants, such as allowing them to record their daily behaviors or administering an established laboratory task for the assessment of actual PEBs.

## Figures and Tables

**Figure 1 ijerph-19-15431-f001:**
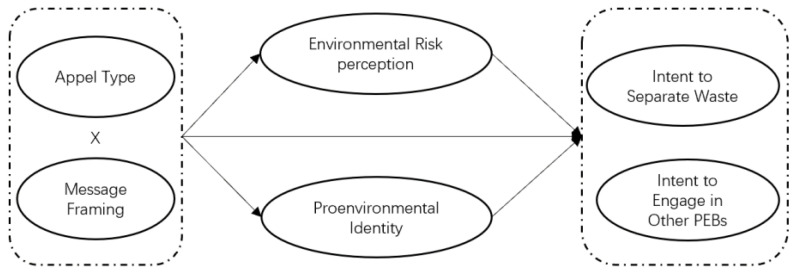
The study’s model.

**Figure 2 ijerph-19-15431-f002:**
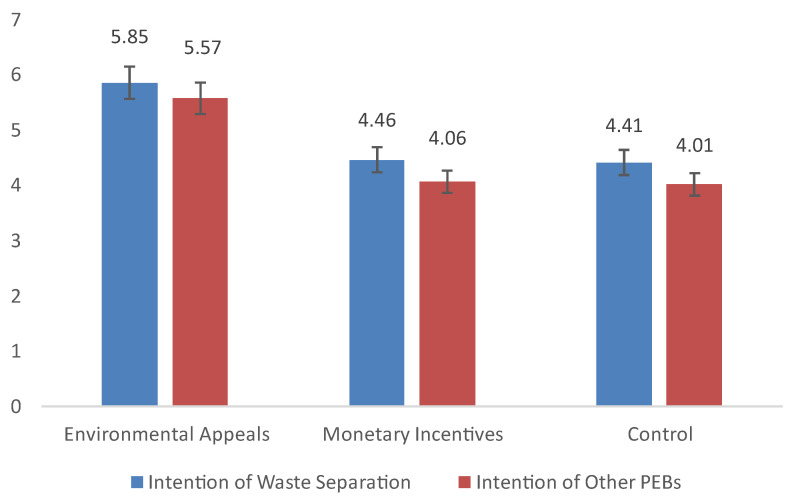
Mean values of intent to separate waste and perform other PEBs for each group. Note: *N* = 126. Error bars show 95% confidence intervals.

**Figure 3 ijerph-19-15431-f003:**
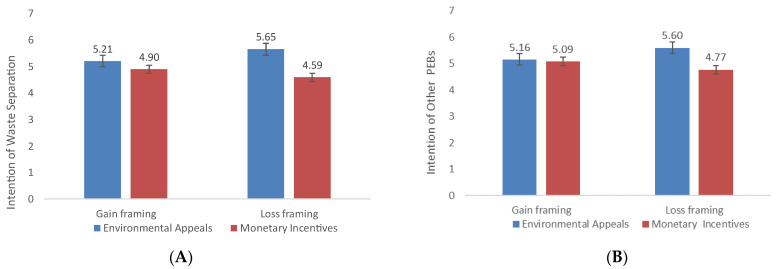
(**A**) Interactive effects of appeal type and message-framing focus on intent to separate waste. (**B**) Interactive effects of appeal type and message-framing focus on intent to engage in other PEBs. Note: *N* = 320. Error bars show 95% confidence intervals.

**Figure 4 ijerph-19-15431-f004:**
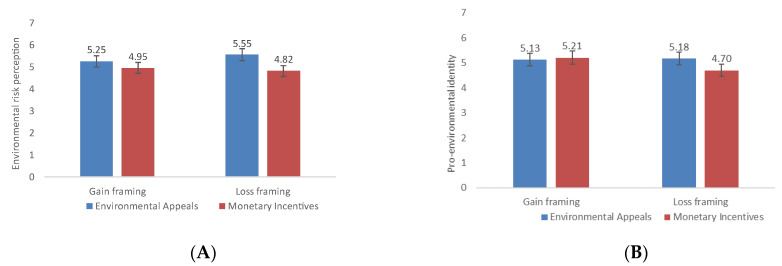
(**A**) Interactive effects of appeal type and message-framing focus on environmental risk perception. (**B**) Interactive effects of appeal type and message-framing focus on pro-environmental identity. Note: *N* = 320. Error bars show 95% confidence intervals.

**Table 1 ijerph-19-15431-t001:** Analysis of the interactive effect of appeal type and message-framing focus on intent to separate waste (other PEBs).

Variable	*df*	*MS*	*F*	*p*	Partial *η*^2^
Dependent variable: Intent to Separate Waste					
Appeal Type	1	37.95	83.48	0.000	0.21
Message Framing	1	0.35	0.77	0.380	0.00
Appeal Type × Message Framing	1	11.03	24.25	0.000	0.07
Error	316	0.46			
Dependent variable: Intention of other PEBs					
Appeal Type	1	16.21	39.05	0.000	0.11
Message Framing	1	0.25	0.60	0.440	0.00
Appeal Type × Message Framing	1	11.63	28.02	0.000	0.08
Error	316	0.42			

Note: *N* = 320.

**Table 2 ijerph-19-15431-t002:** Regression analysis of mediating effects.

**Predictor**	**Mediator = Environmental Risk Perception**		**DV = Intent to Separate Waste**		**DV = Intent to Engage in Other PEBs**	
	** *B* **	**SE** ** _B_ **	** *B* **	**SE** ** _B_ **	** *B* **	**SE** ** _B_ **
Message Framing	0.75 **	0.27	0.03	0.08	0.04	0.08
Appeal Type	0.14	0.27	-	-	-	-
Message Framing × Appeal Type	−0.44 *	0.17	-	-	-	-
Environmental Risk Perception	-	-	0.37 ***	0.05	0.16 **	0.05
**Predictor**	**Mediator = Pro-Environmental Identity**		**DV = Intent to Separate Waste**		**DV = Intent to Engage in Other PEBs**	
	** *B* **	**SE** ** _B_ **	** *B* **	**SE** ** _B_ **	** *B* **	**SE** ** _B_ **
Message Framing	0.59	0.31	0.13	0.08	0.08	0.08
Appeal Type	0.63 *	0.31	-	-	-	-
Message Framing × Appeal Type	−0.55 **	0.20	-	-	-	-
Pro-environmental Identity	-	-	0.29 ***	0.05	0.12 **	0.04

Note: *N* = 320. Appeal type was coded as 1 = environmental appeal and 2 = monetary incentive. Message framing was coded as 1 = gain framing and 2 = loss framing. The presented coefficients are unstandardized estimates. * *p* < 0.05; ** *p* < 0.01; *** *p* < 0.001.

**Table 3 ijerph-19-15431-t003:** Moderated mediation analysis: effects of appeal type and message framing on intent to separate waste and engage in other PEBs.

Variable	Conditional Indirect Effects				Index of Moderated Mediation		
	Moderator	Coefficient	SE	95% CI	Index	SE	95% CI
Mediator: Environmental Risk Perception							
Intent to Separate Waste	EA	0.11	0.04	0.0416, 0.1975	−0.16	0.07	−0.3138, −0.0313
	MI	−0.05	0.05	−0.1617, 0.0551			
Intent to Engage in Other PEBs	EA	0.05	0.02	0.0125, 0.1042	−0.07	0.04	−0.1707, −0.0103
	MI	−0.02	0.03	−0.0860, 0.0205			
Mediator: Pro-environmental Identity							
Intent to Separate Waste	EA	0.01	0.04	−0.0585, 0.1040	−0.16	0.07	−0.3218, −0.0427
	MI	−0.15	0.05	−0.2609, −0.0557			
Intent to Engage in Other PEBs	EA	0.01	0.02	−0.0243, 0.0518	−0.07	0.04	−0.1718, −0.0070
	MI	−0.06	0.03	−0.1431, −0.0096			

Note: *N* = 320. EA and MI indicate the environmental appeal and monetary incentives, respectively. The appeal type was coded as 1 = environmental appeal and 2 = monetary incentive. Message framing was coded as 1 = gain framing, and 2 = loss framing. CI = confidence interval.

## Data Availability

The original contributions presented in the study are included in the article, and further inquiries can be directed to the corresponding author.

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
