# Peer review of "Influence of Appeal Type and Message Framing on Residents’ Intent to Engage in Pro-Environmental Behavior"

_ijerph, 2022, doi:10.3390/ijerph192315431_

Round 1

Reviewer 1 Report

Overall the topic "Influence of Appeal Type and Message Framing on Residents’ Intent to Engage in Pro-environmental Behavior" is timely and well executed with surprising results. I have the following suggestions to support the quality of the work: 

Introduction

1. The first paragraph approach is inadequate, considering a global perspective, and the situation in other regions shall be delineated in the introduction background supported with facts about the adversities of environmental pollution.

2. Similarly, pro-environmental behaviors such as several PEBS are generally used in scale and should be described to give a more compelling story in the introduction section. Focusing on why PEBs are also necessary to substitute existing behavioral patterns.

Literature Review

Overall very well-written portion; however, I suggest adding some details about the message-consistent effect as described in framing theory.

Materials and Methods section:

1. What factors were used in the selection of participants in the study? What were their social background, profession, education, income level, gender, etc.?

2. Did the authors identify or examine the impact of the above parameters on the study's outcome or control?

3. Procedure: How were the interventions (stimuli Appendix B appeals etc.) manipulated using advertisement? or shown content as new etc.? (recognized source?) I was unable to find, please delineate if this information is not available

Discussion

The most theoretically significant result is the surprising result, "although the main effects of the 13 framing of messages on the intent to separate waste and other PEBs were not significant". This shall be discussed with greater insight using the contextual example as it is not aligned with the theoretical assumptions made in Framing theory (particularly message-consistent effect) and interested in discussing by bringing in the context that may play a critical role. A regional or local example can be delineated in this section. 

Author Response

Overall the topic "Influence of Appeal Type and Message Framing on Residents’ Intent to Engage in Pro-environmental Behavior" is timely and well executed with surprising results. I have the following suggestions to support the quality of the work:

Introduction

1.The first paragraph approach is inadequate, considering a global perspective, and the situation in other regions shall be delineated in the introduction background supported with facts about the adversities of environmental pollution.

Thanks a lot for your kind comments. We have rewritten the introduction and added some new data and information as following:

“The development of urbanization and industrialization has had many negative impacts on ecosystems and human health, most of which stem from the imbalance between human social development and ecological governance [1]. The latest Global Risks Report released by the World Economic Forum 2022 pointed out that in the next 5–10 years, environmental risks such as climate action failure, extreme weather, and biodiversity loss rank as the top three most severe risks [2]. Serious ecological crises have aroused widespread public concern and prompted researchers to investigate the mechanisms of individual proenvironmental behaviors (PEBs) to provide practical strategies for improving environmental conditions [3]…”

For details, please see the first three paragraphs of the introduction (modified version, Pp.1-2)

  1. Similarly, pro-environmental behaviors such as several PEBS are generally used in scale and should be described to give a more compelling story in the introduction section. Focusing on why PEBs are also necessary to substitute existing behavioral patterns.

Thanks a lot for your kind comments. We have rewritten the introduction and added some new data and information as following:

“PEBs are generally referred to as “environment protective” or environmentally friendly behavior and involve human activities performed with the aim of protecting the environment or preventing environmental degradation [4], for example, green travel, green consumption, and waste separation. As a major form of public participation in environmental governance, PEBs can significantly improve environmental quality while also providing residents with a sense of well-being and life satisfaction [5]. Whereas the specific PEBs vary widely in terms of residents’ effort, relative financial cost, knowledge, and a range of other factors, some research has shown that there may be some consistency across individuals' different PEBs, and one PEB may affect the other [6,7]. In addition, the likelihood that an individual will adopt one or more non-targeted PEBs depends on the mode of intervention for the target PEB [8,9]…“

Please see the first three paragraphs of the introduction (revised manuscript, Pp. 1-2).

Literature Review

Overall very well-written portion; however, I suggest adding some details about the message-consistent effect as described in framing theory.

Thanks a lot for your kind comments. We have added some details about the message-consistent effect in 2.2 as following:

“Message framing is a type of linguistic expression involving the sharing information with similar meanings through different information encoding methods, such as diction and tone, thus affecting the processing and decoding of content by the audience, resulting in different cognitions and judgments [46]. Framing theory posits that the manner in which information is presented influences the individual responses. As a message design strategy, framing draws attention to certain aspects of information while simultaneously drawing attention away from other aspects. As a consequence of this presentational difference, individuals’ attitudes toward the topic become more aligned with framed information, leading to more message-consistent effects, referred to as a framing effect [47]. Framing theory predicts that processing fluency will mediate the relationship between language frame and message-consistent outcomes because the probability of drawing on framed beliefs through accessibility can be amplified by processing fluency or the ease with which something is processed [48]. In addition, when a message is judged as relevant, it is more likely to lead to message-consistent attitudes and behavior changes [49,50] because it is more likely to be remembered [51].

Gain-loss framing in environmental communication is a common equivalency frame that focuses on differences in how information is presented [47]. Gain-framed messages highlight the positive consequences of behaving in a certain manner. Conversely, loss-framed messages emphasize the risk of not behaving in a certain manner [52]…“

Please see the first two paragraphs of 2.2 (revised manuscript, line 199-211, Pp.4-5).

Materials and Methods section:

1.What factors were used in the selection of participants in the study? What were their social background, profession, education, income level, gender, etc.?

Thanks a lot for your kind comments. We selected the sample randomly in all 3 studies and collected demographic information about them. We are so sorry that we did not present the findings in the original manuscript and we have now supplemented the information about education, income level, and gender (We are sorry that we did not survey the profession of the participants).

Please see the red fonts in the modified version for details and examine line 387-391(Pp.8, study 1), line 453-457(Pp.10, study 2), and line 555-558(Pp.12, study 3).

  1. Did the authors identify or examine the impact of the above parameters on the study's outcome or control?

Thanks a lot for your kind comments. We set these demographic factors as control variables in our study and controlled them in each model. We are so sorry that we did not describe these in the original manuscript and we have supplemented the interpretation in the revised manuscript. Please see the red fonts in the modified version for details and examine line 488-489(Pp.10, study 2) and line 584-586(Pp.13, study 3).

  1. Procedure: How were the interventions (stimuli Appendix B appeals etc.) manipulated using advertisement? or shown content as new etc.? (recognized source?) I was unable to find, please delineate if this information is not available

Thanks a lot for your kind comments. In study 2, the participants came from online recruitment, so we set the manipulation materials as advertisement slogans; The participants in Study 3 came from the offline community, so we set the manipulation materials as public welfare advertisements. We are so sorry for presenting only textual information in the original manuscript. We have supplemented the complete experimental materials in Appendix B.

Discussion

The most theoretically significant result is the surprising result, "although the main effects of the 13 framing of messages on the intent to separate waste and other PEBs were not significant". This shall be discussed with greater insight using the contextual example as it is not aligned with the theoretical assumptions made in Framing theory (particularly message-consistent effect) and interested in discussing by bringing in the context that may play a critical role. A regional or local example can be delineated in this section.

Thanks a lot for your kind comments. We have reviewed a large literature and found that findings on framing research in environmental contexts have not been as consistent. So we have revised the interpretation as following:

“Findings on framing research in environmental contexts have been inconsistent. Some researchers have found a stronger impact of loss-framing on the consumption of green goods or philanthropy than on gain-framing. However, recent research has found that the effects of message framing on food waste prevention intentions [83] and reducing air conditioning use in university dormitories [84] are not significant. Early research also found that positive frames produced more favorable attitudes toward curbside recycling [84,85] and water conservation [85]. In addition, appeals containing both positively and negatively framed statements have been shown to increase recycling behavior [86]. In this study, we found that, although message framing had an influence, the influence differed depending on the appeal type. This difference may have been related to the research context. Although green product consumption and charitable activities are also PEBs, the range of PEBs in this study that we measured was broader. Framing effects occur through processing fluency and applicability, both of which lead to greater frame-consistent perceptions [50]. In addition, framing literature indicates that the involvement level of the message recipient and the interaction of depth of processing are important factors that moderate the persuasiveness of message frames [62]. Among the previous studies that found loss framing to be more effective for PEBs, most experimental materials were related to specific green advertising information. High participant involvement can rapidly form certain advertising attitudes and purchase intentions that may influence message framing. In contrast, because of the vigorous promotion of waste classification by the Chinese government, participants were familiar with the experimental materials, which means that the fluency and applicability of the information were at normal levels. However, participants were not highly engaged in this topic (although we implemented a few measures to improve involvement). Topic involvement refers to the importance and relevance of how an individual perceives a topic, which positively affects their motivation to process information [87]. Therefore, we observed no significant differences in the main effects of message-framing. This result is consistent with that of a previous study, which reported that the loss framework was effective only if it combined specific information and situational factors of participation [61]. Factors such as issue salience (low vs. high) [85], recommended behavior (conservation/taking less vs. recycling/doing more), and the beneficiary of the behavior’s effects (current vs. future generations) [88] affect the persuasive effect of framing. Therefore, there is still room for further research on message framing, and there is an intrinsic relationship between framing design and research context that needs to be explored…”

Please see the red fonts in the modified version for details and examine line 816-849(Pp.18-19).

Reviewer 2 Report

This manuscript will be a useful addition to the literature. However, significant revisions are required in order to achieve this.

The penultimate paragraph of the introduction does not clearly explain the theoretical significance and practical value of the article, and it still needs to be supplemented.

The format of the paper needs to be rechecked, such as: the title of the second subsection is wrongly numbered, the 403rd line is incorrect, etc.

Please give more supporting evidence on whether the results of interviewing 80 samples in the study1 are reasonable for designing the questionnaire.

The results of the 16-sample were used as a measure of PEB's willingness, and the rationality of the questionnaire entry design needed to be supported by more theoretical or authoritative standards.

The method of the study2 lacked a description of the implementation and differentiation of different interventions. There is no specific description of what the intervention environment appeal is, what the intervention of monetary incentives is, how the monetary incentives are implemented, the size of the incentives, and how the incentives are issued.

And Whether the sample size of all experiments meets the need for statistical power.

More information about the experiment should be added.

Author Response

This manuscript will be a useful addition to the literature. However, significant revisions are required in order to achieve this.

1.The penultimate paragraph of the introduction does not clearly explain the theoretical significance and practical value of the article, and it still needs to be supplemented.

Thanks a lot for your kind comments. We have supplemented the theoretical significance and practical value of the article in the introduction as following:

"Overall, our work is a pioneering study on the relationship between information interventions, perceived risk, pro-environmental identity, and environmental responsibility and offers theoretical and applied insights. First, although numerous studies have examined appeal type or message framing on PEBs, such as green consumption [8,20,23], recycling [26,33], and energy savings [34], scholars have not paid sufficient attention to the interactive effect between appeal type and message framing. With this study, we are the first to explore the effectiveness of different types of appeals in motivating self-reported intent to perform a broad range of PEBs from the perspective of matching appeal type and message framing. Second, we investigated the mediating role of environmental risk perception and pro-environmental identity in this process and developed a comprehensive framework. Third, most earlier research focuses on cases from Western Europe, North America, and Australia and has mostly focused on the effects of interventions in energy efficiency on other PEBs. This study aims to advance the spillover literature by examining the effects of waste separation information interventions on Chinese residents‘ intentions to perform other PEBs. Finally, our study provides important theoretical support for the selection and optimization of intervention strategies. "

Please see the red fonts in the revised manuscript for details (line 110-125, Pp.3).

2.The format of the paper needs to be rechecked, such as: the title of the second subsection is wrongly numbered, the 403rd line is incorrect, etc.

Thanks a lot for your kind comments. We have rechecked the format of the paper and revised the error in the original line 403 as following, " In the second phase, the environmental appeal and monetary incentive groups completed a questionnaire ..."

Please see the red fonts in the revised manuscript for details (line 480, Pp.10).

3.Please give more supporting evidence on whether the results of interviewing 80 samples in the study1 are reasonable for designing the questionnaire.

Thanks a lot for your kind comments. In study 1, we asked the participants to evaluate the four environmental motives for performing waste separation behaviors on a seven-point scale. Our aim with the questionnaire was to identify environmentally appealing motivators for subsequent experiments. Second, we randomly selected and assigned 16 participants to two groups to conduct focus group interviews around the open theme of how they understood PEBs. Our aim with the focus group interviews was to improve the scale used for measuring the intent to perform of other PEBs.

4.The results of the 16-sample were used as a measure of PEB's willingness, and the rationality of the questionnaire entry design needed to be supported by more theoretical or authoritative standards.

Thanks a lot for your kind comments. Based on the coding results of the focus group interviews, we referenced the measurement items used by Lanzini and Thogersen (2014) and Xu et al. (2018) and applied 11 items to measure the intent to engaged in other PEBs. Subsequently, we submitted the scale for examination by three scholars in the field of environmental protection who agreed with the items.

Lanzini and Thogersen (2014)’s questionnaire contained 23 PEBs, which comprise the main macro-categories used in most PEB studies (Xu et al.,2018). These behaviors can be summarized into 11 categories, including green purchasing, recycling, public transport, biking, carpooling, switching off lights when exiting room as last person, turn off water when brushing teeth, turn off water in shower, printing documents on both sides, reading documents on screen, and volunteering. Xu et al.'s (2018) scale is popular in China, and the content of the questions is based on Lanzini and Thogersen 's (2014) and Thogersen and þlander's (2003) measurement systems. The questionnaire contained 25 PEBs. These behaviors can be summarized into 7 categories, including household waste separation, domestic electricity/water conservation, green shopping, travelling, support for environmental policies, and willingness to engage in environmental citizenship actions.

We integrated the measurement items of these studies, and reserved items such as green purchasing, green travelling, domestic electricity/water conservation and supporting environmental protection policies. Items such as " printing documents on both sides " and " reading documents on screen " were deleted because most Chinese people have no office and seldom print or read documents on screen. "Willingness to engage in environmental citizenship actions " is modified to " participate in various environmental protection activities " (in the group interviews, many respondents said that they did not understand the concept of citizenship behavior). We added two items, "do not use disposable tableware when dining out" and "bring your own toiletries when traveling". In addition, the "recycling" is divided into three items, “actively recycling and using idle items”, “bring own shopping bag when shopping”, and “reuse plastic bags”, because from the interview and our life experience, these 5 PEBs are exactly the things that many residents think should be done, but it is difficult to adhere to. Consequently, 11 items were summarized to measure other PEB willingness.

5.The method of the study2 lacked a description of the implementation and differentiation of different interventions. There is no specific description of what the intervention environment appeal is, what the intervention of monetary incentives is, how the monetary incentives are implemented, the size of the incentives, and how the incentives are issued.

Thanks a lot for your kind comments. We are so sorry for the lack of a description of the implementation and differentiation of different interventions in study 2. We have revised the interpretation as " The environmental appeal group viewed the material as, “Proper waste separation can prevent harmful substances from entering water sources….” The monetary incentive group viewed the material as, “If you correctly separate waste, you can accumulate points in exchange for eggs, soy sauce….".and supplemented the tips in Appendix B which explain how the monetary incentives are implemented. We are so sorry for presenting only textual information in the original manuscript.

Please see the red fonts in the revised manuscript for details (line 463-465, Pp.10).

6.And Whether the sample size of all experiments meets the need for statistical power.

Thanks a lot for your kind comments.

We predicted the sample size by G*power (Faul et al., 2007) before the experiment, and for the one-way ANOVA applied to Study 2, at α = 0.05 and moderately higher effect (f = 0.3), the total sample size required to predict a level of power = 85% was at least126, and we actually recruited 126 valid participants (42 in each experimental condition).

For the two-way ANOVA applied to Study 3, at α = 0.05 and moderately higher effect (f = 0.3), the total sample size required to predict a level of power = 85% was at least 141 (main effects) and 102 (interaction effects), and we actually recruited 320 valid participants (80 in each experimental condition).

Reference:

Faul, F., Erdfelder, E., Lang, A.-G. and Buchner, A. (2007). G*Power 3: A flexible statistical power analysis program for the social, behavioral, and biomedical sciences. Behav Res Methods. 39, 175-191.doi: 10.3758/bf03193146

7.More information about the experiment should be added.

Thanks a lot for your kind comments. We have supplemented some experimental details in study 2 and 3 and revised the original presentation which was not clear. We are so sorry for presenting only textual information in the original manuscript. We have supplemented the complete experimental materials in Appendix B.

Study 2: " …The environmental appeal group viewed the material as, “Proper waste separation can prevent harmful substances from entering water sources….” The monetary incentive group viewed the material as, “If you correctly separate waste, you can accumulate points in exchange for eggs, soy sauce….”

Study 3: " …the first part of the situational stimulus material used gain- or loss-framed messages to describe the results of waste separation behaviors owing to monetary or environmental factors, such as “Through the correct separation of household waste…”; “If household waste is not properly separated…”; “Of household waste, 30–40% can be recycled…”; and “Of domestic waste, 30–40% can be recycled….” Detailed information is presented in Appendix B. Next, to enhance their involvement, we asked participants in each of the two gain framing groups to write down the environmental benefits and monetary benefits they would receive from separating waste. Participants in each of the two loss framing groups were asked to write down the environmental losses or monetary losses they might experience from not participating in waste separation…"

Please see Appendix B and the red fonts in the modified version for details (line 463-465, Pp.10; line 569-576, Pp.12).

Reviewer 3 Report

This research investigates the effect of message appeal (environmental vs monetary incentives) and message framing (loss vs gain) on intention waste separation and intention of other PEBs as well as mediator roles of pro-environmental identity and environmental risk perception un such relationships.

The chosen subject is valuable for the scholars and practitioners, however there are major issues need to be addressed to be warranted for publication:

1)                  Introduction should highlight the importance of the research and provides information on possible contributions.

2)                  Environmental risk perception and pro-environmental identity as research variables should be introduced in the introduction.

3)                  Arguments for mediators which are environmental risk perception and pro-environmental behaviour should be split.

4)                  I would prefer to start introduction to position sustainable problem as a global phenomenon supported with a recent reference, then example could be China.

5)                  After H4, graphical representation of the model should be added to increase understandability.

6)                  Arguments in p.2. rows 51-52 needs to be supported from the literature..(e.g. Ref)

7)                  It would be better to a few lines of explanation what environmental appeal and monetary incentives means in the research context under the title of 2.1.

8)                  Arguments in p.4., rows 154-156 seems redundant, as H2 is about the effect of loss (vs gain) frames on intention....

9)                  H3a and H3b should be rewritten to explain in comparison to what, environmental appeal with loss framing  or monetary incentives with gain framing is more effective.

10)              The study should clarify whether one of the DV is Intention of other PEBs (this is mentioned on the instrument) or engagement of other PEBS (this is mentioned in the hypotheses)

11)              What the abbreviation of PEB refers to in the first sentence of second paragraph? Please expand abbreviation when it is first time used always.

Author Response

This research investigates the effect of message appeal (environmental vs monetary incentives) and message framing (loss vs gain) on intention waste separation and intention of other PEBs as well as mediator roles of pro-environmental identity and environmental risk perception un such relationships.

The chosen subject is valuable for the scholars and practitioners, however there are major issues need to be addressed to be warranted for publication:

1) Introduction should highlight the importance of the research and provides information on possible contributions.

Thanks a lot for your kind comments. We have supplemented the theoretical significance and practical value of the article in the introduction as following:

“Overall, our work is a pioneering study on the relationship between information interventions, perceived risk, pro-environmental identity, and environmental responsibility and offers theoretical and applied insights. First, although numerous studies have examined appeal type or message framing on PEBs, such as green consumption [8,20,23], recycling [26,33], and energy savings [34], scholars have not paid sufficient attention to the interactive effect between appeal type and message framing. With this study, we are the first to explore the effectiveness of different types of appeals in motivating self-reported intent to perform a broad range of PEBs from the perspective of matching appeal type and message framing. Second, we investigated the mediating role of environmental risk perception and pro-environmental identity in this process and developed a comprehensive framework. Third, most earlier research focuses on cases from Western Europe, North America, and Australia and has mostly focused on the effects of interventions in energy efficiency on other PEBs. This study aims to advance the spillover literature by examining the effects of waste separation information interventions on Chinese residents‘ intentions to perform other PEBs. Finally, our study provides important theoretical support for the selection and optimization of intervention strategies.”

Please see the red fonts in the revised version for details (line 110-125, Pp.3).

2) Environmental risk perception and pro-environmental identity as research variables should be introduced in the introduction.

Thanks a lot for your kind comments. We have supplemented the introduction of environmental risk perception and pro-environmental identity as following:

“Environmental risk perception is one of the prerequisites for the public to choose and implement environmental behaviors [27], and different types of message framing elicit emotional responses, thereby influencing risk perception [28]. Some studies have shown that the public's perception of environmental risk for a particular environmental issue, such as air pollution, water pollution, and climate change, plays a significant role in promoting PEBs [14,15]. Furthermore, many scholars believe that the relationship between pro-environmental identity and the adoption of PEBs is well-established because social identities are important predictors of behavior when a behavior is understood to be relevant to that identity [29,30]. Some studies have shown that information interventions that frame the benefits of a PEB in monetary versus moral terms can forge a new or stronger pro-environmental identity that is powerful enough to trigger the adoption of secondary behaviors, such as increased public preferences for environmentally friendly products, environmental intentions, and policy support [31,32]. Therefore, we examined the role of environmental risk perception and pro-environmental identity on the influence of appeal type and message framing on residents’ intent to engage in PEBs.”

Please see the green fonts in the revised version for details (line 95-109, Pp.2-3).

3) Arguments for mediators which are environmental risk perception and pro-environmental behaviour should be split.

Thanks a lot for your kind comments. We have described these two mediators separately in 2.4 and 2.5

Please see the green fonts in the modified version for details (line 254, Pp.6; line 320,Pp.7).

4) I would prefer to start introduction to position sustainable problem as a global phenomenon supported with a recent reference, then example could be China.

Thanks a lot for your kind comments. We have substantially revised the introduction and supplemented some important data and information from a global perspective, such as " The development of urbanization and industrialization has had many negative impacts on ecosystems and human health, most of which stem from the imbalance between human social development and ecological governance [1]. The latest Global Risks Report released by the World Economic Forum 2022 pointed out that in the next 5–10 years, environmental risks such as climate action failure, extreme weather, and biodiversity loss rank as the top three most severe risks [2]. Serious ecological crises have aroused widespread public concern and prompted researchers to investigate the mechanisms of individual proenvironmental behaviors (PEBs) to provide practical strategies for improving environmental conditions [3]…", and also adding some data on waste separation in China such as "… A report released by the World Bank in 2018 showed that in high-income countries, one-third of the waste is recycled and composted, whereas in low-income countries, the recycling rate is only 4% [10]. China is one of the countries with the highest pressure for municipal domestic waste disposal in the world, with domestic waste production increasing dramatically at a rate of 8–10% per year [11]. However, the recycling rate of domestic waste in China was only 15.6% in 2015…",

Please see the first three paragraphs of the introduction (Pp.1-2).

5) After H4, graphical representation of the model should be added to increase understandability.

Thanks a lot for your kind comments. We have added the graphical model, please see Figure 1(Line 377, Pp.8).

6) Arguments in p.2. rows 51-52 needs to be supported from the literature..(e.g. Ref)

Thanks a lot for your kind comments. We have added the specific literature for every argument as following, “…such as the design of waste-separation policies [14],media [15], social interaction [16], community support and investment in hardware facility [17]…” 

Please see the green fonts in the modified version for details (line68-69, Pp.2).

7) It would be better to a few lines of explanation what environmental appeal and monetary incentives means in the research context under the title of 2.1.

Thanks a lot for your kind comments. We have added the explanation of environmental appeal and monetary incentives under the title of 2.1 as following, “Environmental appeals mainly refer to bringing awareness to the hazards caused by waste pollution and emphasizing to residents the contribution of waste separation to environmental protection, with the aim of fostering awareness of environmental norms and encouraging residents to voluntarily separate waste. Monetary incentives refer to the provision of financial rewards to residents who participate in waste separation or informing them of the fines they may face for not participating in waste separation to increase their motivation to separate [35]. “

Please see the green fonts in the modified version for details (line135-141, Pp.3)

8) Arguments in p.4., rows 154-156 seems redundant, as H2 is about the effect of loss (vs gain) frames on intention....

Thanks a lot for your kind comments. We have deleted that explanation.

9) H3a and H3b should be rewritten to explain in comparison to what, environmental appeal with loss framing or monetary incentives with gain framing is more effective.

Thanks a lot for your kind comments. We have rewritten H3a and H3b as following:

Hypothesis 3a (H3a): The intention to separate waste and other PEBs is stronger when environmental appeals match loss framing rather than gain framing.

Hypothesis 3b (H3b): The intention to separate waste and other PEBs is stronger when monetary incentives match gain framing rather than loss framing.

Please see the green fonts in the modified version for details (line280-293, Pp.6)

10) The study should clarify whether one of the DV is Intention of other PEBs (this is mentioned on the instrument) or engagement of other PEBS (this is mentioned in the hypotheses)

Thanks a lot for your kind comments. We are so sorry for ambiguity caused by the language The previous expressions were "intent to separate waste and engage in other PEBs". We have modified all the interpretation to"the intention to waste separation and other PEBs" in the hypotheses.

Please see the green fonts in the modified version for details (line192-193, Pp.4; line243-244, Pp.5; line315-318, Pp.7; line356-358, Pp.8)

11) What the abbreviation of PEB refers to in the first sentence of second paragraph? Please expand abbreviation when it is first time used always.

Thanks a lot for your kind comments. With the modification made to the introduction section, the first appearance of PEB has changed and we have ensured that we expanded abbreviation when it was first time used.

Round 2

Reviewer 3 Report

I am happy with the amendments on the manuscript. Just some minor points before warranted for publication.

1.       Last sentence of the abstract should be replaced with a one important practical implication.

2.       P.2, row, 47, after that sentence, one specific example should follow, may be from reference 6,7.

3.       It would be better to include in 3a and 3b, in comparison to what?

4.       It would be better to remove or update last sentence of H4a, the reason is as follows, H4 refers to the mediator role of environmental risk perception between IVs and intention, however, that sentence only refers to the first path, not to the path from environmental risk perception to the intention. Same comment for H4b. Alternative to removing that sentence could be addition of “,,”which leads to greater/higher intention bla bla”.

Author Response

  1. Last sentence of the abstract should be replaced with a one important practical implication.

Thanks a lot for your kind comments. We have replaced the last sentence as following: “…we provide important theoretical support for the selection and optimization of waste separation intervention strategies and guidance for persuading residents to engage in more PEBs...”

Please see the red fonts in the revised version for details (line 23-25, Pp.1).

  1. 2, row, 47, after that sentence, one specific example should follow, may be from reference 6,7.

Thanks a lot for your kind comments. We have added two specific examples as following, “…For example, those who reduced their red meat consumption were more likely to make donations to environmental organizations【8】… Evans et al(2013)found that discarded paper recycling was significantly higher than control when participants received environmental information about car-sharing, but was no different from control when they received financial information or received both financial and environmental information[9].”

Please see the red fonts in the revised version for details (line 47-54, Pp.2).

  1. It would be better to include in 3a and 3b, in comparison to what?

Thanks a lot for your kind comments. We have revised H3a and 3b as following:

Hypothesis 3a (H3a): When exposed to environmental appeals, participants faced loss framing information will show stronger intention to separate waste and other PEBs than faced gain framing information.

Hypothesis 3b (H3b): When exposed to monetary incentives, participants faced gain framing information will show stronger intention to separate waste and other PEBs than faced loss framing information.

Please see the red fonts in the revised version for details (line 285-290, Pp.6).

  1. It would be better to remove or update last sentence of H4a, the reason is as follows, H4 refers to the mediator role of environmental risk perception between IVs and intention, however, that sentence only refers to the first path, not to the path from environmental risk perception to the intention. Same comment for H4b. Alternative to removing that sentence could be addition of “,,”which leads to greater/higher intention bla bla”.

Thanks a lot for your kind comments. We have removed last sentences of H4 and H5.

Please see the red fonts in the revised version for details (line 320-322, Pp.7; line358-360, Pp.8).
